# Self-Adaptive Pendulum-Ball Switches for Piezoelectric Synchronous-Extraction Circuits

**DOI:** 10.3390/mi13040532

**Published:** 2022-03-28

**Authors:** Yao Huang, Gang Qin, Weiqun Liu

**Affiliations:** School of Mechanical Engineering, Southwest Jiaotong University, Chengdu 610031, China; yao_huang@tongji.edu.cn (Y.H.); qingangsurui@foxmail.com (G.Q.)

**Keywords:** piezoelectric energy extraction, self-adaptive mechanical switches, phase advance, synchronous switching circuit

## Abstract

Electronic synchronous switches are usually used to enhance the performance of piezoelectric energy-extraction circuits, but the electronic components leading to additional power consumption are not desired for energy extraction. In view of the advantage of mechanical switches without power consumption, this article proposed a synchronous-switch circuit which can adapt to the amplitude of a cantilever-beam-vibration generator with less energy loss. This mechanical switch consists of two pendulum balls and two buffer springs. This switch mechanism can automatically adapt to the cantilever-displacement amplitude, control the closing and opening of switches with the decrease in phase advance angle, and increase in energy-extraction efficiency. Different from previous adaptive mechanical switches, this unique pendulum-ball mechanism can not only reduce the weight and volume of the generator to improve the energy density, but can also simply adjust the pendulum length to achieve better harvesting performance. It is verified experimentally that the adaptive mechanical switch can close and open automatically under different cantilever amplitudes and excitation frequencies; the results show that the optimal power of the proposed circuit can reach 4.2 times that of the standard circuit. In order to further optimize the adaptive mechanical switch, the parameters of the swing-ball mechanism affecting harvesting performance is analyzed.

## 1. Introduction

With more and more low-power electronic components, energy harvesting is a key technology for the power supply for wireless sensor network. Piezoelectric materials with high energy density and low cost are widely used in vibration-energy harvesting [1,2]. In addition, a piezoelectric-energy generator can supply power to wireless sensors [3] and wearable devices [4], which not only saves energy but also greatly improves the service life of the device.

A vibration-energy generator is mainly composed of mechanical power generation and an energy-extraction circuit. To improve the energy-extraction efficiency of the generator, on the one hand, the energy-generator structure can be optimized to increase the bandwidth [5,6,7,8]; on the other hand, the energy-extraction circuit can be improved, such as SSHI (Synchronized Switching Harvesting on an Inductor) [9,10], DSSH (Double SSH) [11], SECE (Synchronous Electric-Charge Extraction) [12], OSECE (Optimized SECE) [13], Tunable SECE [14], etc. According to the research, the energy-extraction efficiency of these circuits is higher than that of the traditionally standard energy-extraction circuit, but their performance needs further study. The typical energy-extraction circuits mentioned above are shown in Figure 1. Among them, a simple one is the standard circuit which compromises a rectifier and a filtering capacitance. However, the generator’s charge is not completely transferred to the energy-storage element; thus, the performance is poor in low-electromechanical-coupling cases. To fully utilize the generated charge and enhance the harvested power, the SSHI technique mainly makes some improvements on the basis of the standard energy-extraction circuit, which includes two types of realizations: serial SSHI and parallel SSHI, as shown in Figure 1c,d. For the serial SSHI, the synchronous switch and the inductor are connected in series between the piezoelectric element and the rectifier. While the parallel SSHI consists of adding up a switching device in parallel with the piezoelectric element, as shown in Figure 1d, this switching device is composed of a switch and an inductance. The switch is almost always open, except when a displacement extremum occurs. At this instant, the switch is closed. The capacitance of the piezoelectric element and the inductance then constitutes an oscillator. The switch is kept closed until the voltage on the piezoelectric element has been reversed.

Most of the circuits depend on the synchronous switches to close and open near the voltage peak to increase the extraction efficiency, while common synchronous switches include electronic circuit breakers [15,16,17,18,19,20], speed-control switches [21] and mechanical switches [22,23,24,25,26]. Among the above switches, electronic circuit breakers and speed-control switches are controlled by electronic devices, which requires a certain starting voltage and consumes part of the power, while the starting voltage of the mechanical switch is low and does not consume extra energy. In fact, most of the current mechanical switches [22,23,24] are designed with fixed electrodes; when the amplitude is constantly changing, the mechanical switches can hardly work normally, which is not conducive to energy extraction. In addition, there is a mechanical switch [26] design of two-degree-of-freedom mechanism, which can only work normally near the natural frequency and has poor environmental adaptability. Another two kinds of adaptive mechanical switches [27,28] have been designed, which can work normally under most conditions, and the energy extraction efficiency is greatly improved, while the extraction efficiency is also limited by the large phase advance when the mechanical switch is closed. Recently, Liu et al. [29] presented a mechatronic switch using the snubber structure of viscous materials; the harvesting performance of the harvester is sensitive to the parameters of the auxiliary oscillator or snubber structure, especially the stiffness, but the harvesting performance is slightly affected by the auxiliary mass. Obviously, such a complex structure makes it difficult to adjust and optimize the energy harvester. Therefore, an adaptive mechatronic switch with easy adjustment of structural parameters is expected. Furthermore, though there have been extensive researches on mechanical switches, most of them focus on how to automatically adapt to the displacement amplitude of the harvester. On this basis, there are few investigations on lighter and smaller mechanical switches to improve the energy density of the harvester.

In this paper, a novel lightweight SSHI-PBMS (SSHI with Pendulum-Ball Mechanical Switch) circuit is proposed as seen in Figure 2. The proposed design is composed of a main piezoelectric generator and two auxiliary oscillators. Specifically, the cantilever beam with PZT and adjustable mass is used as the generator to convert vibration energy into electrical energy, which contributes to evaluating the performance of mechanical switches. With regard to the auxiliary oscillator, it includes a cycloid with adjustable length and a ball with adjustable mass. This adaptive mechanical switch is composed of two pendulum-ball mechanisms and two buffer springs fixed at the end of the cantilever beam. The equivalent stiffness of the moving electrode is very low so that the phase advance angle is very small; therefore, the energy-extraction efficiency is further improved. Moreover, the stiffness of the auxiliary oscillator depends on the pendulum length, which indicates that the pendulum length can be simply adjusted to improve the performance of the auxiliary oscillator, so as to ensure that the energy harvester can work efficiently. The proposed mechanical switch can adapt to the peak displacement of the cantilever and work normally in a wide frequency range without additional energy consumed. The experiments show that the proposed circuit can work normally under different amplitudes and excitation frequencies; the maximum harvested power of the proposed circuit is 4.2 times that of the standard circuit and the harvested power is also higher than other adaptive mechanical-switch circuits. Obviously, the harvested energy of the proposed circuit can fully meet the power demand of most wireless sensors.

## 2. Modeling and Principle

The proposed design is based on a cantilever piezoelectric-energy generator with pendulum balls as fixed electrodes, which are composed of guide wires and light metal balls. Buffer springs are set on both sides of the inertia mass as the center electrodes and the two pendulum balls serve as moving electrodes. These electrodes form two mechanical switches and are used in the SSHI-PBMS circuit, as shown in Figure 2. The buffer springs can reduce the rigid collision between the ball and the mass to improve the stability of the pendulum-ball mechanism. It is worth noting that the pendulum ball makes only contact with the buffer spring, and the stiffness of the thin guide wire in the horizontal direction can be ignored. When the pendulum ball swings, the stiffness of the pendulum-ball mechanism is very low, which is different from other adaptive mechanical switches.

Figure 3 shows the simplified model of the generator structure: the cantilever generator and two pendulum mechanisms are represented by three typical spring-mass-damping systems. The proposed pendulum switch has a lower damping, *μ*_2_. Moreover, a special piezoelectric element is added in the simplified model of the beam to include the electromechanical-coupling effects. It is noted that the electric charge is mainly converted from the first mode of the beam with PZT. The charge induced by higher modes is neglected. The piezoelectric element with piezoelectric-force factor *α* is viewed as an equivalent current source *i* and an intrinsic capacitor *C*_0_. Therefore, the dynamic equations of the system can be written as follows:
(1)$$M_{1}{\ddot{x}}_{1}+K_{1}x_{1}+\mu_{1}{\dot{x}}_{1}+F_{1}(x_{1},x_{2})+F_{2}(x_{1},x_{3})+\alpha V_{p}=M_{1}\gamma$$
(2)$$i=\alpha {\dot{x}}_{1}-C_{0}{\dot{V}}_{p}$$
(3)$$M_{2}{\ddot{x}}_{2}+K_{2}x_{2}+\mu_{2}{\dot{x}}_{2}-F_{1}(x_{1},x_{2})=M_{2}\gamma$$
(4)$$M_{2}{\ddot{x}}_{3}+K_{2}x_{3}+\mu_{2}{\dot{x}}_{3}-F_{2}(x_{1},x_{3})=M_{2}\gamma$$
in which *γ* is the excitation acceleration; *x*_1_, *x*_2_ and *x*_3_ are the displacements of the cantilever beam and the two pendulum balls, respectively. In the initial state *x*_1_ = *x*_2_ = *x*_3_ = 0, *M*_1_ and *M*_2_ are the equivalent masses of the cantilever beam and the pendulum ball. *K*_1_ and *K*_2_ are the stiffness of the generator and the pendulum, where *K*_2_ = *M*_2_gtan(arcsin(*x*_2_/*L*))/*x*_2_. *L* is the length of the cycloid; *μ*_1_ and *μ*_2_ are the damping coefficients; *F*_1_ and *F*_2_ are the interaction forces between the cantilever and the pendulum balls, respectively. Since the pendulum length *L* is much larger than *x*_2_, the *x*_2_ can be regarded as approximately equal to the vibration amplitude of the pendulum ball *A_m_*. From the equation for stiffness *K*_2_, it can be seen that there are two methods to reduce *K*_2_: one is to increase the length of the pendulum *L* and the other is to reduce the mass of the pendulum ball *M*_2_. Consequently, from the above analysis it is not difficult to find that the value of stiffness *K*_2_ is affected by the pendulum length *L* and the pendulum ball mass *M*_2_, which is conducive to improving the performance of synchronous switches. Since *F*_1_ and *F*_2_ only exist when the cantilever is in contact with the pendulum balls, the corresponding expressions can be derived:(5)$$F_{1}(x_{1},x_{2})=[K_{3}(x_{1}-x_{2})+\mu_{3}({\dot{x}}_{1}-{\dot{x}}_{2})]\times H(x_{1}-x_{2})$$
(6)$$F_{2}(x_{1},x_{3})=[K_{3}(x_{1}-x_{3})+\mu_{3}({\dot{x}}_{1}-{\dot{x}}_{3})]\times H(-x_{1}+x_{3})$$
where *K*_3_ and *μ*_3_ represent the stiffness and damping coefficient of the buffer spring, respectively. *H*(*x*) is the contact condition, only when *x*_1_ > *x*_2_ > 0, *H*(*x*) = 1 and S_1_ is closed, other moments *H*(*x*) = 0 and S_1_ is open. Similarly, when *x*_1_ < *x*_3_ < 0, *H*(*x*) = 1 and S_2_ is closed, other moments *H*(*x*) = 0 and S_2_ is open.

The working principle of the adaptive mechanical switch is shown in Figure 4. As shown in Figure 4a, in the initial state, *x*_1_ = *x*_2_ = *x*_3_ = 0, with the cantilever moving towards the positive direction, the peak value *x*_1_ = *A_m_* is reached, the displacement of the pendulum ball in the positive direction *x*_2_ = *A_m_*, as shown in Figure 4b. Then, the cantilever moves to the negative direction until the negative peak value *x*_1_ = −*A_m_*, and the pendulum ball displacement *x*_3_ = −*A_m_* in the negative direction, as shown in Figure 4c. In other words, the cycloid stiffness in the horizontal direction is low, so that the natural frequency of the auxiliary vibrator is kept at a low level to adapt to the frequency of the surrounding environment. In this way, the displacement of the pendulum ball in the positive direction is basically unchanged *x*_2_ ≈ *A_m_*. When the cantilever moves to the positive direction again, it only needs to lose a small part of kinetic energy to make the pendulum ball in the positive direction reach the positive peak displacement again, as shown in Figure 4d; the motion of the pendulum ball in the negative direction is similar to that of the pendulum ball in the positive direction.

When the system is stable, the cantilever and the pendulum ball repeat the above processes (Figure 4c,d). The moving electrode of the adaptive mechanical switch contacts the central electrode alternately near the displacement extremum of the cantilever, and the switches S_1_ and S_2_ close and open respectively near the displacement extremum of the cantilever. When the displacement of the beam increases or decreases, the ball is able to rely on the kinetic energy or gravitational potential energy of the beam to achieve new balance and stay near the peak displacement of the beam; thus, the adaptive mechanical switch can automatically track the displacement amplitude of the beam and control the synchronous switch to close near the displacement extremum without consuming additional electric energy. There is no doubt that in the harvesting process, there would be a small amount of energy loss due to the collision between the auxiliary oscillator and the cantilever beam, but the pendulum-ball switch designed in this paper has less energy loss because of its light weight.

## 3. Experiments and Results

According to the above principle, a piezoelectric-energy generator is designed and manufactured, which is composed of one stainless-steel beam with the size of (100 × 20 × 1) mm and two piezoelectric patches with sizes of (30 × 20 × 0.4) mm. The two piezoelectric patches, with capacitances of 2.7 × 10^−8^ F, are respectively pasted on the two surfaces near the fixed end of the cantilever beam, and the piezoelectric patches are connected in parallel, as shown in Figure 5. A mass block is fixed near the free end of the cantilever beam. To avoid the rigid collision between the mass block and the moving electrodes, the foam is pasted on both sides of the mass block as support, and the buffer spring is formed by covering a layer of arc-conductive tape on the foam block. The stiffness of the buffer spring 200 N/m is much lower than that of the cantilever beam 1032 N/m. The combination of mass block, foam block and buffer spring is used as the central electrode. Two pendulum-ball mechanisms are vertically installed on the bracket, and the pendulum ball is adjusted to be located on both sides of the central electrode of the cantilever. Considering the low rigidity and conductivity of the moving electrode, the cycloid is made of conductive tape, and the pendulum ball is made of a stainless-steel ball. In order to connect the pendulum ball with the cycloid, a layer of conductive tape is covered on the pendulum ball.

This experimental set-up is installed on the shaker (The Modal Shop©, 2075e-ht) which is driven by a signal generator (RIGOL©, DS1022Z) to generate the excitation signal; the laser sensor (SUNX©, hl-c203be) obtains the displacement of the cantilever; the piezoelectric voltage and cantilever displacement are displayed and saved through the oscilloscope (Tektronix©, MDO3024). The acceleration sensor (PCB©, M352C68) is mounted on the shaker to obtain the practical excitation acceleration. Based on the standard diode-rectifier circuit, a parallel SSHI circuit is composed of an inductor and a mechanical switch. The parameters in the system are shown in Table 1. The natural frequency of the cantilever beam measured by manual sweeping is 49.4 Hz, which is close to the theoretical value of 50 Hz.

To verify the performance of the designed structure, the first step is to maintain the excitation frequency of 50 Hz and obtain the waveforms of the piezoelectric-patch voltages and cantilever displacement under three displacement amplitudes, as shown in Figure 6. It can be seen that the adaptive mechanical switch can work normally at low amplitude and high amplitude. In the figure, dotted lines represent the reverse position of the voltage waveform of the piezoelectric patches, that is, the closed position of the mechanical switch. In addition, the closed position is close to the displacement peak of the cantilever beam. At the same time, through the derived model, the displacement-simulation waveforms of the cantilever beam and the pendulum ball are obtained, and the closed position of the mechanical switch is marked with the circle. Obviously, when *x*_1_ > *x*_2_, S_1_ is closed and disconnected after the peak displacement, and when *x*_1_ < *x*_3_, S_2_ is closed. In Figure 6, it can be observed that the results of the experiment and simulation match well, and the switches S_1_ and S_2_ can automatically open and close near the displacement peak of the cantilever beam. Careful observation shows that the phase advance angle of the mechanical switch increases slightly with the increase in the amplitude of the cantilever beam; due to the increase in the pendulum amplitude of the ball, the stiffness *K*_2_ increases and the restoring force of the pendulum ball increases. On account of the certain mass and rigidity of the pendulum mechanism, the closing times of the mechanical switches S_1_ and S_2_ are always earlier than the peak displacements of the cantilever beam, which is inevitable. It is not difficult to see that the curve part in the voltage matches with the sine displacement in [29]. However, in this paper in Figure 6, the curve part of the voltage has the opposite shape to the displacement. In fact, the positive and negative of the voltage value is not directly related to the positive and negative of the displacement value. The positive and negative of the displacement value is determined by the installation position of the laser sensor and the cantilever beam. However, the positive and negative of the piezoelectric voltage value is related to the polarization direction of the piezoelectric plates and the wiring sequence of two interfaces of the interface circuit.

In addition, it is verified that the amplitude of the cantilever beam is *A_m_* = 1 mm under different excitation frequencies. As shown in Figure 7, there are three waveforms: (a) 48 Hz, (b) 50 Hz, (c) 52 Hz. The results show that the adaptive mechanical switch can work stably at different excitation frequencies, and the experimental and simulation results match well. However, it can be found that with the increase in the excitation frequency, the phase advance angle slightly decreases. This is because the higher the excitation frequency, the less time it takes for the switch to close twice continuously. Finally, the smaller the distance of the pendulum-ball restoration, the closer the closed position of the switch is to the displacement extreme of the cantilever beam.

To further illustrate the performance advantages of the circuit with the adaptive mechanical switch, the following experiments are carried out, keeping the amplitude of the cantilever beam as 1 mm, selecting the excitation frequency as 45 Hz, 48 Hz, 50 Hz, 52 Hz and 55 Hz, respectively, and the load-resistance value changes between 20 kΩ and 2 MΩ to measure the load power. For comparison, the standard energy-extraction circuit is tested under the same conditions. Figure 8 shows that the load resistance at which maximum power is obtained in the SSHI-PBMS circuit is higher than that of the standard circuit, and the power of the proposed circuit is higher than that of the standard circuit under the optimal resistance. In Figure 8a, when the excitation frequency changes from 45 Hz to 55 Hz, the optimal load of the SSHI-PBMS circuit decreases, while the corresponding maximum load power increases from 2.38 mW to 3.34 mW. In Figure 8b, the optimal load of the standard circuit also decreases with the increase in the excitation frequency, and the maximum load power increases from 0.77 mW to 0.92 mW. Compared with the standard circuit, the load power of the SSHI-PBMS circuit is always higher, which also shows that the proposed self-adaptive mechanical switch can work normally in different cantilever-displacement amplitudes.

Similarly, the excitation frequency is kept at 50 Hz and the amplitudes of the cantilever beam are 0.5 mm, 0.75 mm, 1 mm, 1.25 mm and 1.5 mm, respectively; the load powers of the two circuits are measured, as shown in Figure 9. Obviously, the load power of the SSHI-PBMS circuit is higher than that of the standard circuit under all conditions, and their power ratio is always more than 240%. When the cantilever-beam amplitude is 0.5 mm, the optimal power of the SSHI-PBMS circuit is 0.79 mW and the optimal power of the standard circuit is 0.33 mW, while the cantilever beam amplitude is 1.5 mm, the optimal power of the SSHI-PBMS circuit is 6.76 mW and the optimal power of the standard circuit is 1.63 mw, and the power improvement is significantly increased. More importantly, as can be seen from Figure 8 and Figure 9, the SSHI-PBMS circuit works well over a wide range of frequencies and amplitudes, with a significant increase in load power compared to the standard circuit.

For comparison, the maximum power of the two circuits is shown separately. The load power increases with the increase in excitation frequency; as shown in Figure 10a, the SSHI-PBMS circuit is always higher than the standard circuit, and the power of the SSHI-PBMS circuit is 3.6 times that of the standard circuit under the excitation frequency of 55 Hz. Figure 10b shows the relationship between the optimal power of the two circuits and the cantilever amplitude. Obviously, the power increases rapidly with the increase in the cantilever amplitude, and the load power of the SSHI-PBMS circuit is always better than that of the standard circuit. The higher the amplitude, the more obvious the power increase. In addition, the power of the proposed circuit is 4.2 times that of the standard circuit under the displacement amplitude of 1.5 mm. At the same time, the optimal powers of other adaptive mechanical-switch circuits are also much higher than those of the standard circuit, but its power is slightly lower than that of the proposed circuit under different conditions, which further indicates that the energy-extraction efficiency of the proposed adaptive mechanical switch is higher.

As mentioned earlier, it is not difficult to find that the length of the cycloid can affect the stiffness *K*_2_. Therefore, the following experiments are carried out to analyze the influence of pendulum length on harvested power: keeping the excitation frequency at 50 Hz, the acceleration at 2 m s^−2^ and the load resistance at 800 KΩ. Selecting different cycloid lengths to measure the power of the load resistance, as shown in Figure 11a, it can be found that with the increase in cycloid length, the load power corresponding to the two circuits increases slowly, which is because after the cycloid becomes longer the kinetic-energy loss of the system decreases, so the load power of the circuit increases. Thus, relying on the adjustment of the pendulum length of the mechanical switch, a larger power output can be obtained without the adjustment of more complex structural parameters, which undoubtedly provides better convenience for the application of the energy harvester in more practical environments. It should be noted that it is difficult to ensure that the installation position of the cycloid is consistent all the time during each measurement, which leads to some measurement errors. In the next chapter, the influence of cycloid length on the system will be further discussed through the model.

Meanwhile, the case of constant excitation acceleration is analyzed. As shown in Figure 11b, the excitation-frequency acceleration is 0.2 g, and the excitation frequency continuously changes from 45 Hz to 55 Hz. Obviously, the displacement for 45 Hz or 55 Hz is much lower than 50 Hz in Figure 11b, while the power output is similar, at 45 Hz or 55 Hz and 50 Hz in Figure 10a. In Figure 10a, the optimal resistance value under 50 Hz excitation is selected, and the acceleration is adjusted to keep the amplitude of the cantilever beam at 1 mm under all excitation frequencies. When the displacement amplitude remains constant, more energy is generated at the same time under the excitation of higher frequency, so the power output in Figure 10a increases slowly from 45 Hz to 50 Hz. Meanwhile, the acceleration is constant at 0.2 g in Figure 11b, so there is a large displacement only near the resonance frequency. The displacement amplitude of the cantilever decreases slightly when the mobile electrode is introduced disconnected to the circuit, which shows that the designed self-adaptive mechanical switch has little impact on the system. However, when the designed switch is connected to the circuit, the displacement amplitude decreases significantly, which indicates that the system damping increases, and more kinetic energy is converted into electrical energy. In order to ensure the consistency, keeping the mechanical switch connected to the standard circuit, the displacement amplitude is slightly lower than that of the unconnected circuit, but significantly higher than that of the self-adaptive mechanical-switch circuit, which also shows that the energy extraction efficiency of the standard circuit is not high, while the designed switch can effectively improve the extraction efficiency. From the anterior analysis, the designed adaptive mechanical-switch structure has a small equivalent stiffness and small damping, so the pendulum ball can be kept near the peak displacement of the cantilever beam with little energy.

Through the experimental study of constant cantilever amplitude and excitation frequency, the adaptive performance of the self-adaptive mechanical switch is verified, which proves that the structure can work efficiently in a wide range of excitation frequencies and cantilever amplitudes. At the same time, in comparison of the same research on the standard circuit, it is proved that the load power of the proposed circuit is significantly higher than that of the standard circuit, and the higher the excitation frequency and acceleration, the more obvious the advantages of the SSHI-PBMS circuit. Moreover, compared with the previous adaptive mechanical switch [29] whose performance was greatly affected by the stiffness and damping of the auxiliary oscillator and buffer structure, the proposed mechanical-switch structure is simpler and easy to adjust to improve the harvesting performance by changing the pendulum length.

## 4. Further Discussions

The advantages of the proposed adaptive mechanical switch have been verified by experiments. In order to further improve the performance of the pendulum-ball mechanism, we conduct an in-depth study on the system parameters. In fact, SSHI technology can improve the performance of the circuit mainly depending on the synchronous switch in the circuit, and the closing time of the synchronous switch directly affects the efficiency of the circuit. Therefore, this paper uses the phase advance angle of the closing time of the mechanical switch to evaluate the performance of the proposed mechanical switch.

Using the simulation model, the main system parameters of the pendulum structure are studied and analyzed. Five excitation frequencies (40 Hz, 45 Hz, 50 Hz, 55 Hz and 60 Hz) are selected for analysis, as shown in Figure 12. Firstly, the length of the cycloid is analyzed; as shown in Figure 12a, with the decrease in cycloid length *L*, the phase advance angle decreases rapidly and then slowly. The reason is because when the amplitude is constant, the longer cycloid length *L* leads to the smaller equivalent stiffness *k*_2_, and the restoration of the pendulum ball is slower, thus the position of the second closing of the synchronous switch is closer to the displacement peak, resulting in the decrease of the phase advance angle. The smaller the phase advance angle, the lower the energy loss when the cantilever beam contacts the pendulum ball, which is conducive to improving the load power. Figure 12b shows the effect of the pendulum-ball mass on the phase advance angle. Obviously, the pendulum-ball mass increases, and the phase advance angle almost increases linearly. This is because the pendulum-ball mass directly affects the equivalent stiffness *K*_2_; the larger the pendulum-ball mass, the greater the equivalent stiffness *K*_2_ and the faster the pendulum ball recovers. Therefore, choosing a smaller pendulum-ball mass *M*_2_ can make the mechanical switch close in time to realize charge retention, unlike the previous mechanical switch whose mass adjustment has little influence on the harvesting performance [29]. In this way, the power density of the energy harvester can be significantly improved, and the energy loss caused by the pendulum-ball switch with smaller mass can be greatly reduced. Figure 12a,b shows that the higher the excitation frequency, the smaller the corresponding phase advance angle, which is caused by the decrease in the interval time between two consecutive closures of the switch. Meanwhile, from the comparison between Figure 12a,b, the gradual curve in Figure 12b implies that the pendulum-ball mass *M*_2_ has a smaller effect on the phase advance angle than the pendulum length. Consequently, adjusting the pendulum length *L* to change the cycloid stiffness *K*_2_ is also a good choice to improve the performance of the pendulum-ball switch.

For further quantitative analysis, according to the experimental results under different frequencies and constant amplitudes, the power comparison between the SSHI-PBMS circuit and the standard circuit is summarized in Table 2. The results at 40 Hz, 50 Hz and 55 Hz are obtained with the constant amplitude 1 mm, while the results at 0.5 mm, 1 mm and 1.5 mm are obtained with the frequency 50 Hz. It can be found that for both the standard circuit and the SSHI-PBMS circuit, the power increases with the increase in frequency or displacement amplitude, with other conditions remaining unchanged. However, in general, the power ratio of the SSHI-PBMS circuit is between 200% and 500% compared with the standard circuit in all cases, which implies the advantage of the proposed mechanical switch in energy harvesting.

From the foresaid study related to the influence of the load connected in the electrical circuit, the proposed generator with SSHI-PBMS shows huge potentiality for providing sustainable and clean power sources for sensors. Figure 13 presents a possible application scheme with the proposed SSHI-PBMS generator. By putting the piezoelectric generator with the SSHI-PBMS circuit under the vibration environment, the vibrations can be converted into electricity by the generator. The harvested energy is then supplied to a wireless node through a power-management unit. This node integrates the desired sensors (vibration, temperature etc.) for acquiring the status of the device required to be monitored and sends the information to the coordinator through the wireless link. Therefore, the desired status can be monitored in real time.

## 5. Conclusions

In this article, a novel adaptive mechanical-switch circuit adopting a simple harmonic-motion structure as a moving electrode is proposed. The mechanical switch can automatically adapt to the displacement amplitude of the cantilever beam, and it can close and open near the displacement amplitude of the cantilever beam. Due to the equivalent stiffness of the pendulum mechanism being lower than other adaptive mechanical switches, the closed position of the mechanical switch is closer to the peak displacement, and the energy-extraction efficiency is higher. The performance of the adaptive mechanical switch has been verified by experiments, and it can work normally in a wide range of cantilever amplitudes and excitation frequencies. In the experiment, the maximum power of the SSHI-PBMS circuit was 6.76 mW at the cantilever amplitude of 1.5 mm, while the maximum power of the standard circuit was 1.63 mW, and the maximum power of SSHI was 4.2 times that of standard circuit.

For further optimization, the main parameters of the pendulum mechanism have been studied. It is found that the mass of the pendulum ball determines the equivalent stiffness, and the equivalent stiffness directly affects the phase advance angle of the mechanical switch; thus, choosing a lower pendulum-ball mass is conducive to improving the performance of the mechanical switch, which is not only helpful to reduce relevant energy loss, but also contributes to the portability and integration of energy harvesters in specific environmental applications. Similarly, under the same conditions, increasing the length of the cycloid can also reduce the equivalent stiffness of the pendulum mechanism. Therefore, a better adaptive mechanical switch with better harvesting performance can be designed by selecting appropriate parameters. In short, two key factors in the design of self-adaptive pendulum-ball switches are that the pendulum length *L* of the auxiliary oscillator should be longer, but too long a cycloid is disadvantageous for the miniaturization design of the energy harvester; whereas the pendulum-ball mass *M_2_* should be as small as possible.

The performance of the adaptive mechanical switch proposed in this paper has obvious advantages over the previous self-adaptive mechanical switches. Thanks to the unique lightweight pendulum-ball structure, the energy density of the generator is further improved with less energy loss. Although part of the excitation conditions is analyzed in this article, the proposed mechanical switch can adapt to more excitation conditions and expand the application by appropriate designs.

## Figures and Tables

**Figure 1 micromachines-13-00532-f001:**
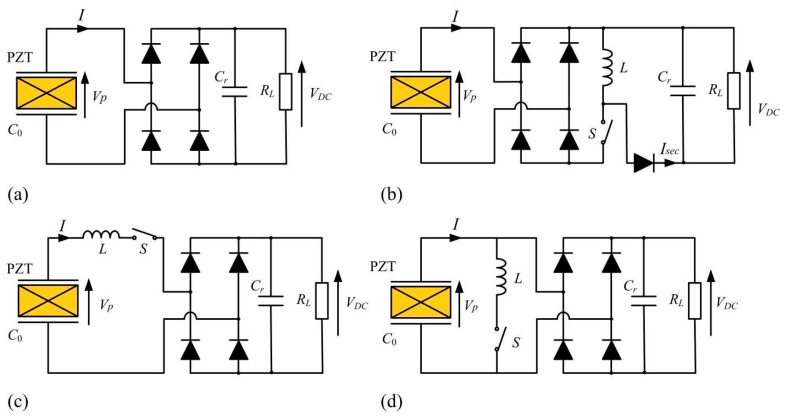
Energy-extraction circuits: (**a**) standard circuit; (**b**) SECE circuit; (**c**) serial SSHI circuit; (**d**) parallel SSHI circuit.

**Figure 2 micromachines-13-00532-f002:**
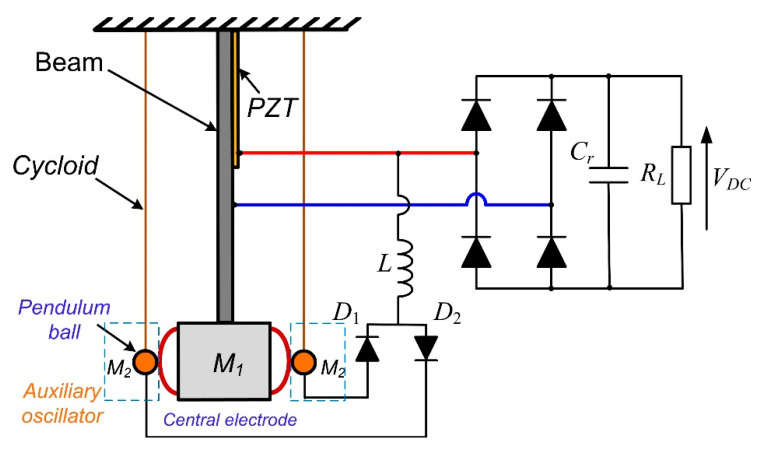
Piezoelectric generator with the proposed adaptive mechanical switch.

**Figure 3 micromachines-13-00532-f003:**
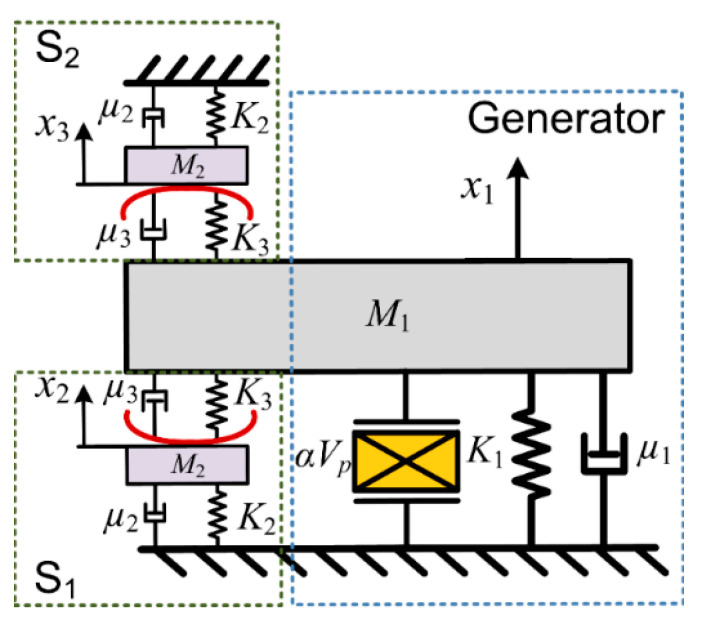
Model for the generator with the proposed mechanical-switch structure.

**Figure 4 micromachines-13-00532-f004:**
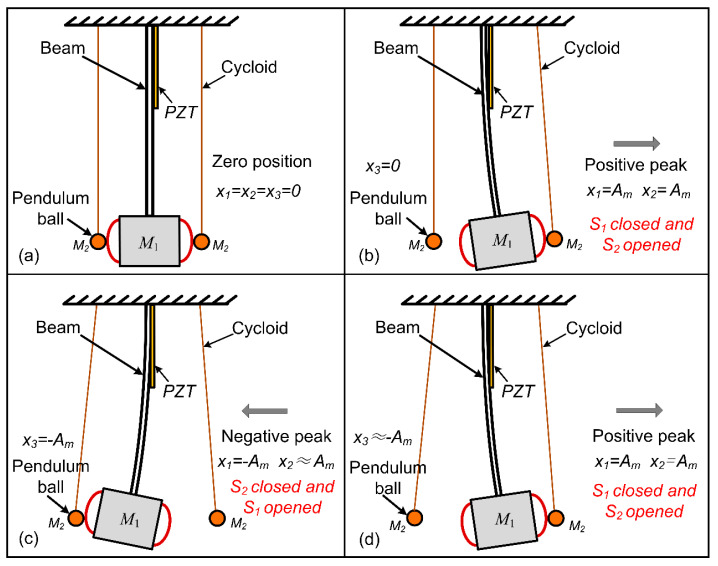
Working principle of the proposed self-adaptive mechanical switch by several snapshots of the first period: (**a**) initial status; (**b**) positive peak; (**c**) negative peak; (**d**) repeated positive peaks.

**Figure 5 micromachines-13-00532-f005:**
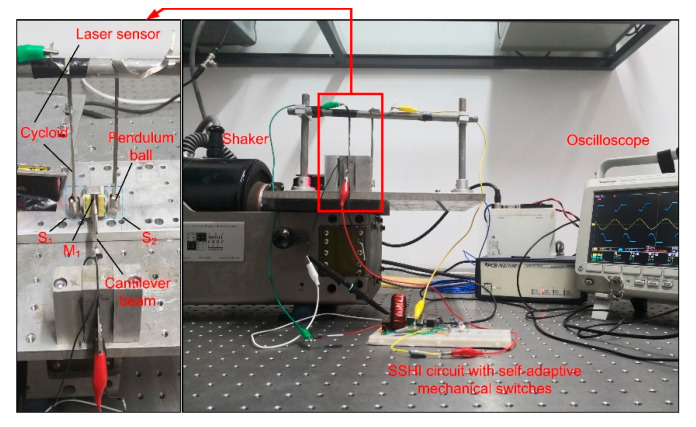
Experimental set-up and prototype.

**Figure 6 micromachines-13-00532-f006:**
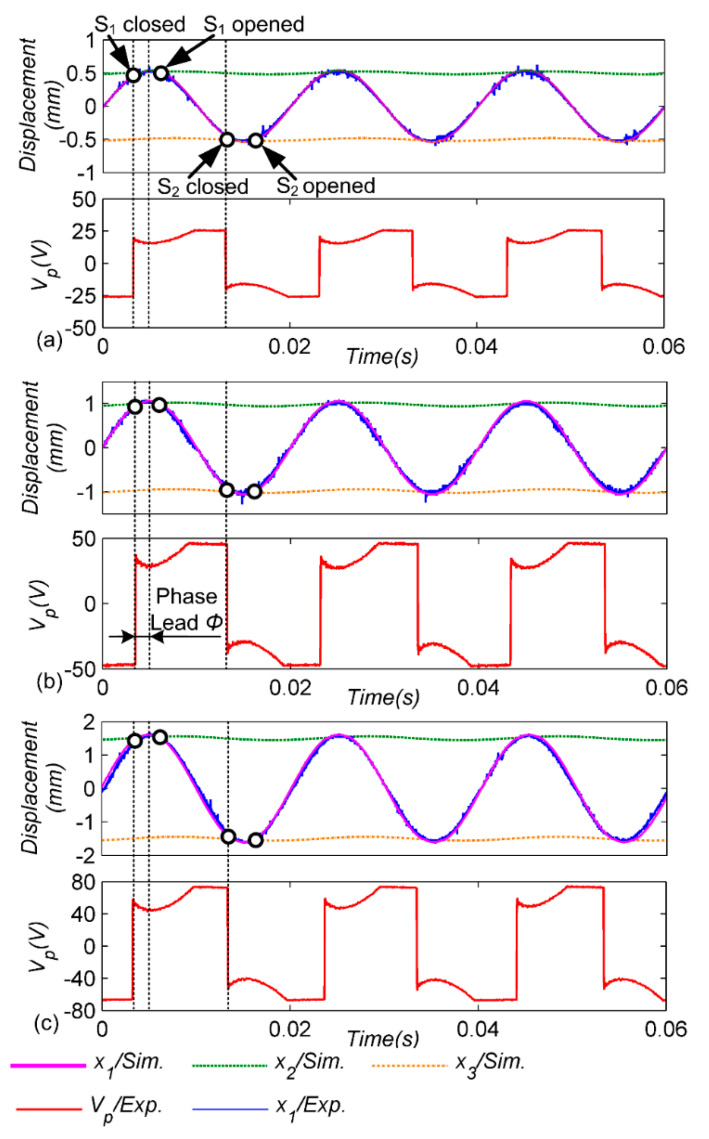
Experimental and simulated waveforms for different vibration amplitudes at a constant frequency *f* = 50 Hz: (**a**) *A_m_* = 0.5 mm; (**b**) *A_m_* = 1 mm; (**c**) *A_m_* = 1.5 mm.

**Figure 7 micromachines-13-00532-f007:**
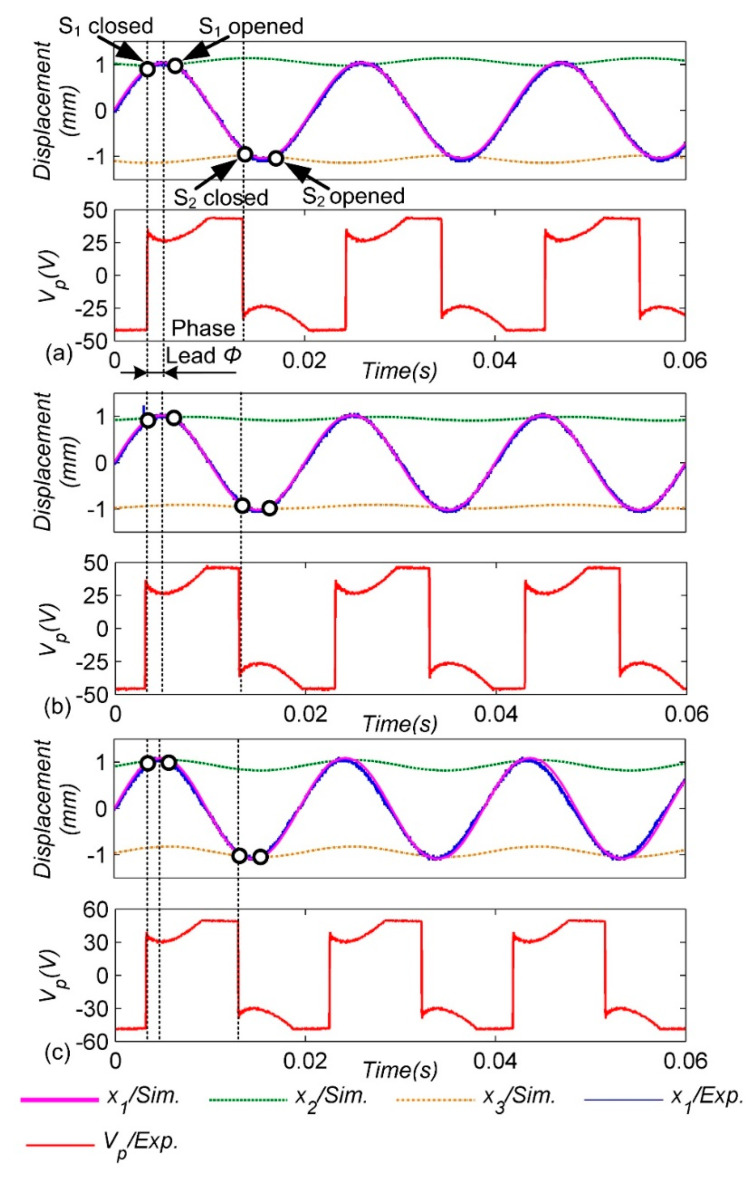
Experimental and simulated waveforms for different excitation frequencies with a constant amplitude *A_m_* = 1 mm: (**a**) 48 Hz; (**b**) 50 Hz; (**c**) 52 Hz.

**Figure 8 micromachines-13-00532-f008:**
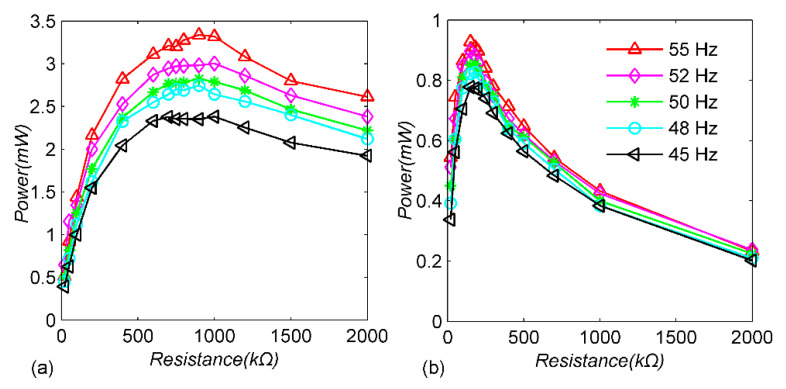
Experimental harvested power for different excitation frequencies and load values in the constant-displacement case: (**a**) SSHI-PBMS circuit; (**b**) standard circuit.

**Figure 9 micromachines-13-00532-f009:**
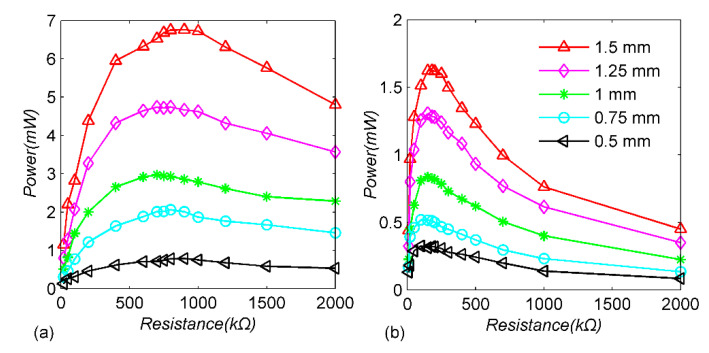
Experimental harvested power for different amplitudes and load values in the constant-excitation-frequency case: (**a**) SSHI-PBMS circuit; (**b**) standard circuit.

**Figure 10 micromachines-13-00532-f010:**
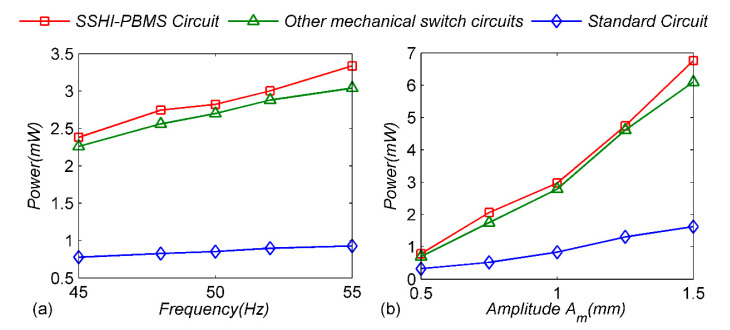
Experimental maximum-harvested-power comparison between SSHI-PBMS circuit and standard circuit for varied displacement frequencies or amplitudes: (**a**) constant amplitude 1 mm; (**b**) constant frequency 50 Hz.

**Figure 11 micromachines-13-00532-f011:**
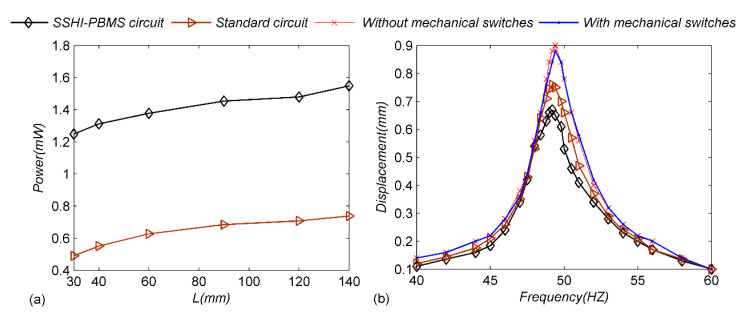
(**a**) Power responses of the SSHI-PBMS circuit and the standard circuit; (**b**) Displacement responses of the generator for different situations.

**Figure 12 micromachines-13-00532-f012:**
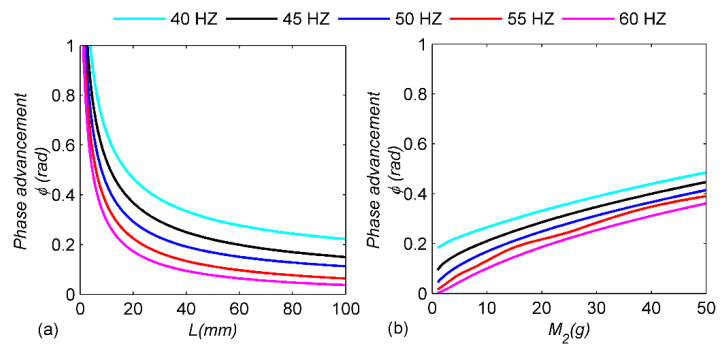
Parameter analysis of the pendulum mechanism: (**a**) the length of the cycloid *L*; (**b**) the mass of the pendulum ball *M*_2_.

**Figure 13 micromachines-13-00532-f013:**
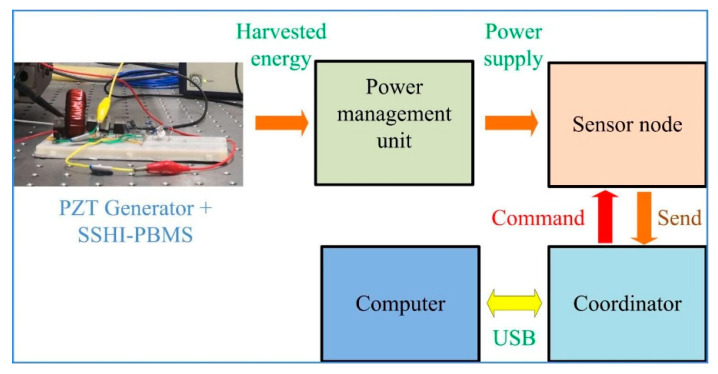
A potential application scenario with the proposed SSHI-PBMS generator.

**Table 1 micromachines-13-00532-t001:** Parameters for the proposed prototype.

Symbol	Value	Symbol	Value
Pendulum length *L* (mm)	100	Piezoelectric-force factor *α* (N V^−1^)	0.0003
Equivalent mass of beam *M*_1_ (kg)	0.0105	Equivalent mass of pendulum ball *M*_2_ (kg)	0.004
Equivalent damping of beam *μ*_1_ (N m^−1^ s)	0.03	Equivalent damping of buffer spring *μ*_3_ (N m^−1^ s)	0.5
Equivalent stiffness of beam *K*_1_ (N m^−1^)	1032	Equivalent stiffness of buffer spring *K*_3_ (N m^−1^)	200
Piezoelectric capacitance *C*_0_ (F)	2.7 × 10^−8^		

**Table 2 micromachines-13-00532-t002:** Power comparison between SSHI-PBMS circuit and standard circuit.

		Standard Circuit (mW)	SSHI-PBMS Circuit (mW)	Power Ratio
Frequency (Hz)	40	0.78	2.38	306%
	50	0.85	2.82	331%
	55	0.93	3.34	360%
Amplitude (mm)	0.5	0.33	0.79	239%
	1	0.84	2.97	354%
	1.5	1.63	6.76	415%

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
