# Peer review of "Self-Adaptive Pendulum-Ball Switches for Piezoelectric Synchronous-Extraction Circuits"

_micromachines, 2022, doi:10.3390/mi13040532_

Round 1
Reviewer 1 Report
In this work the authors propose a pendulum based mechanical switching mechanism for better energy extraction from a piezoelectric energy harvester. The focus of the paper is just on the switching mechanism and not the energy harvester or power circuit. Here are the reviewers comments:
- Please provide figures for what is "standard circuit", "SSHI circuit" and "other mechanical switching circuit". What is "auxiliary oscillator"? Clearly label in figure 1 and clearly describe "cycloid/ball pendulum with adjustable length and mass" and "cantilever beam with PZT and adjustable mass".
- The piezoelectric part of the model is not properly described. It is shown in figure 2 and included in Equation 1 and 2, but not described in the manuscript at all. The description is essential because later in the results (Figure 5,6), the simulation of this model is shown.
- Figure 5 and Figure 6 are same. In figure 6 the frequency should change and amplitude should be 1mm for all cases. Thus Figure 6 should be corrected.
- The equation K2=M2g*tan[(arcsin(x2/L))/x2] is important and should be properly derived. It describes how M2 and L effect K2 which in turn effect the switching. eg, increasing L reduces K2; reducing M2 reduces K2; since L>>x2 => x2~Am. The authors repeatedly mention "the component force of the pendulum ball gravity in the horizontal direction", which makes no sense because gravity is always in vertical direction. Instead the behaviors can be simply explained from the equation for K2.
- In Ref [29], figure 3, the curve part in the voltage matches with the sine displacement. But in this paper in Figure 5, curve part of voltage has opposite shape than displacement. Why?
- Figure 10b shows displacement for 45Hz/55Hz is much lower than 50Hz, but Figure 9a shows the power output is similar at 45Hz/55Hz and 50Hz. How?
Reviewer 2 Report
Dear authors,
Thank you for this work, which proposes a synchronous switching circuit that can be adapted to the amplitude of the cantilever beam vibration generator, for obtaining a higher energy efficiency. The results obtained are promising.
To improve your work, here are some suggestions:
- To specify in the Introduction what is the meaning of the term SSHI.
- The way of writing the dimensions could be: for steel beam (100x20x1) mm; for patches (30x20x0.4) mm.
- In Fig. 4 - for clarity of image, the components may be written in red.
- In the description of the experimental system, it is useful to mention some data and for PTZ / SSHI / electrical circuit.
- In Table 1 - the names of the parameters should also be specified. Also for C0.
- In Fig. 6 - the image does not correspond to the title and the explanations given.
-In Fig. 5 and Fig. 6 - explain how the voltage performance of about 50 V was obtained. The measurement was made without connecting a load resistor?
- For the study related to the influence of the load connected in the electrical circuit, it is welcome to introduce a new figure, with the scheme of the electromechanical system considered.
- A brief description of the standard energy extraction circuit should also be given. What are the technical features compared with proposed system?
- The phrase “Fig. 7 shows that the optimal resistance of the proposed SSHIPBMS circuit is higher ... ”should be modified (load resistance at which maximum power is obtained in the circuit ..)
- A space must be left between the value and the unit of measurement (0,5 mm, 0,75 mm, 1 mm, 1,25 mm…)
- The results obtained should be centralized in a table of results.
Round 2
Reviewer 1 Report
The authors have sufficiently addressed my concerns in the revised manuscript. Thus I recommend the revised manuscript for publication.